# Quantitative Assessment of Ciliary Ultrastructure with the Use of Automatic Analysis: PCD Quant

**DOI:** 10.3390/diagnostics11081363

**Published:** 2021-07-29

**Authors:** Andrea Felšöová, Tibor Sloboda, Lukáš Hudec, Miroslav Koblížek, Petr Pohunek, Vendula Martinů, Žofia Varényiová, Simona Kadlecová, Jiří Uhlík

**Affiliations:** 1Department of Histology and Embryology, Second Faculty of Medicine, Charles University, V Úvalu 84, 150 06 Prague 5, Czech Republic; simona.kadlecova@lfmotol.cuni.cz (S.K.); jiri.uhlik@lfmotol.cuni.cz (J.U.); 2Clinical and Transplant Pathology Centre, Institute for Clinical and Experimental Medicine, Vídeňská 1958, 140 21 Prague 4, Czech Republic; 3Institute of Computer Engineering and Applied Informatics, Faculty of Informatics and Information Technologies, Slovak University of Technology, Ilkovicova 2, 84216 Bratislava, Slovakia; slobodaapl@gmail.com (T.S.); lukas.hudec@stuba.sk (L.H.); 4Department of Pathology and Molecular Medicine, Second Faculty of Medicine, Charles University and Motol University Hospital, V Úvalu 84, 150 06 Prague 5, Czech Republic; koblizekm@centrum.cz; 5Department of Paediatrics, Second Faculty of Medicine, Charles University and Motol University Hospital, V Úvalu 84, 150 06 Prague 5, Czech Republic; petr.pohunek@lfmotol.cuni.cz (P.P.); vendy.martinu@gmail.com (V.M.); zofia.varenyiova@fnmotol.cz (Ž.V.)

**Keywords:** cilia, primary ciliary dyskinesia, secondary ciliary dyskinesia, automatic analysis

## Abstract

The ciliary ultrastructure can be damaged in various situations. Such changes include primary defects found in primary ciliary dyskinesia (PCD) and secondary defects developing in secondary ciliary dyskinesia (SCD). PCD is a genetic disease resulting from impaired ciliary motility causing chronic disease of the respiratory tract. SCD is an acquired condition that can be caused, for example, by respiratory infection or exposure to tobacco smoke. The diagnosis of these diseases is a complex process with many diagnostic methods, including the evaluation of ciliary ultrastructure using transmission electron microscopy (the golden standard of examination). Our goal was to create a program capable of automatic quantitative analysis of the ciliary ultrastructure, determining the ratio of primary and secondary defects, as well as analysis of the mutual orientation of cilia in the ciliary border. PCD Quant, a program developed for the automatic quantitative analysis of cilia, cannot yet be used as a stand-alone method for evaluation and provides limited assistance in classifying primary and secondary defect classes and evaluating central pair angle deviations. Nevertheless, we see great potential for the future in automatic analysis of the ciliary ultrastructure.

## 1. Introduction

Motile cilia are apical specializations [1], protrusions of the cell membrane averaging 10 micrometers in length, which contain a complex structure built from cytoskeletal proteins called the axoneme [2]. The axoneme is composed of nine doublets of microtubules deposited in the periphery and one central pair of microtubules connected by a complex of bridging proteins and surrounded by an inner sheath [3]. Peripheral microtubules are connected to the central pair by radial spokes that maintain the complex structure of the axoneme [4] and are interconnected by an N-DRC (nexin dynein regulatory complex), which is highly elastic, allowing freedom and coherence of movement [5]. The peripheral doublets consist of two microtubules, A and B, of which only the A microtubule is formed by a complete number of 13 protofilaments [6]. Microtubule A contains microtubule-associated dynein motor protein complexes that form dynein arms, divided into a group of inner and outer dynein arms [7]. It is the interaction of these dynein arms with the peripheral doublet of microtubules in close proximity that is the basis for the active movement of the cilia [8]. The exact number of peripheral microtubules, their correct connection to the central pair, and correct orientation identical in the ciliary border, as well as the occurrence of connecting stabilization and motor complexes is a prerequisite for the proper function of the cilia and their regular movement. 

The ciliary ultrastructure can be damaged in various situations. Such changes include primary defects found in primary ciliary dyskinesia (PCD) and secondary defects developing in variable conditions that can lead to secondary ciliary dyskinesia (SCD) [9,10]. 

PCD is a heterogeneous genetic disorder that can result in chronic lung disease, rhinosinusitis, recurrent infectious lung disease, bronchiectasis with possible associated hearing loss, infertility, and laterality defects [11]. The basis of the disease is a defect in the motility of the cilia normally ensuring airway clearance by moving and clearing mucus and pathogens [12]. Reduced mucus clearance thus leads to chronic inflammation in the upper and lower airways with possible development of permanent damage [13]. The prevalence of the disease is around 1 in 10,000 live births, the reported tendency of the disease prevalence to increase is probably a consequence of improving diagnostic methods and the expansion of the genetic background [14].

SCD is an acquired condition, which can be caused, for example, by respiratory infection or exposure to tobacco smoke. The deterioration of ciliary movements is usually reversible [15]. The importance of early and accurate diagnosis significantly affects the prognosis of patients and their early inclusion in the treatment process. The diagnosis of PCD is complex and should be carried out in specialized diagnostic centers [16]. PCD is associated with a large number of ultrastructural defects of the ciliary axoneme, most of which can be observed in an electron microscope image and can manifest as characteristic ciliary beat patterns observed in high-speed video microscopy (HSVM) [17]. 

Currently, the evaluation of the ciliary ultrastructure using transmission electron microscopy (TEM) is one of the main methods used to diagnose PCD [18]. This diagnostic method is combined with the currently commonly used HSVM, immunofluorescence and molecular genetic testing [19]. Criteria for TEM diagnostics were recently defined in international consensus guidelines [20]. Thus, the quantitative assessment of the ciliary ultrastructure is an integral component of the TEM examination [21]. To distinguish fine ultrastructural defects more precisely, software that enhance the images of ciliary ultrastructure are currently being developed [22].

We focused on the development of the automatic analysis of TEM images that could quantitatively analyze primary and secondary defects on a large number of available cilia and thus be a useful tool to improve and accelerate diagnosis. We evaluated primary changes detectable in a given magnification (25,000×), such as complex central pair defects, microtubule disorganization and absence of peripheral microtubules (not defects of dynein arms), and secondary changes, such as the presence of free axonemes, compound cilia, edema of the ciliary membrane and the presence of a higher number of peripheral microtubules. 

Since the individual cilia in the respiratory epithelium are thought to beat in a direction perpendicular to the plane through the two central microtubules and move in one direction for proper efficiency, our next goal is the automatic analysis of the mutual orientation of cilia. The ability to cooperate in the cleansing of the respiratory tract and work as a functional unit is determined by the deviation of the central pair angles of the individual cilia [23,24].

## 2. Materials and Methods

Samples of ciliated epithelium were obtained from patients of the Pediatric Department of the Motol University Hospital in Prague, Czech Republic, mainly using nasal brushing. A few samples came from endobronchial biopsy during bronchoscopic examination. Samples were obtained from patients with symptoms suspected of PCD: specific personal or family history, recurrent respiratory infection, bronchiectasis, hearing impairment, or organ laterality defects. Sampling for electron microscopy was preceded by repeated physical examinations, examinations of nasal NO levels in adult patients and children with developed paranasal sinuses capable of performing a velum closure maneuver, and HSVM examinations. The patients or their parents signed the informed consent approved by the Ethical Committee of the Motol University Hospital in Prague (grant project No. NV19-07-00210). Recruitment of patients in this study did not increase the number of collected samples compared to the routine diagnostics. 

Processing of samples for electron microscopy was according to standard procedure [25,26]. Briefly, nasal brushings were fixed with 2.5% glutaraldehyde (Merck, Prague, Czech Republic) overnight, and endobronchial biopsy specimens were fixed with 5% glutaraldehyde for 90 min. Fixed specimens were rinsed and post-fixed by 2% osmium tetroxide (Sigma-Aldrich, Prague, Czech Republic). Nasal brushing suspensions were then centrifuged and solidified by 2% agarose (Sigma-Aldrich, Prague, Czech Republic). All specimens were dehydrated by a graded series of ethanol, cleared in propylene oxide (Sigma-Aldrich, Prague, Czech Republic), and embedded in Durcupan-Epon mixture (Fluca, Prague, Czech Republic). Polymerized tissue blocks were sectioned by the ultramicrotome EM UC6 (Leica Microsystems, Vienna, Austria). Semi-thin sections were stained with toluidine blue and ultrathin sections (70 nm thickness) were put into 200 mesh copper grids and contrasted with uranyl acetate and lead citrate. Electron micrographs at an original magnification of 25.000x were captured by the JEM 1011 TEM (JEOL, Tokyo, Japan) equipped with the CCD camera Morada (Olympus SIS, Münster, Germany).

In our center, we have used manual taxonomic marking and counting of individual cilia for quantitative evaluation of the defects thus far, shown in Figure 1. For manual examination in the NIS Elements AR Imaging Software (Laboratory Imaging, Prague, Czech Republic), we use a taxonomic table in which we grouped the individual primary and secondary defects of the cilia into the defined subgroups according to their significance.

The evaluation of primary defects at a defined magnification (25.000×) was classified into subgroups: a defect of the central pair, whether it is the absence of one or both microtubules from the central pair or the shift of the peripheral microtubule doublet to the center of the axoneme, disorganized arrangement of microtubules and absence of peripheral microtubules. The presence and quality of outer and inner arms were evaluated at a higher magnification, allowing sufficient resolution of at least 50 cross sections of cilia without artifacts created during sample preparation and photography. In the overall assessment, using the NIS Elements program, we followed the latest international guidelines that classify ciliary defects into two classes [20]. Class 1 represents the hallmark defects for PCD, and class 2 indicates PCD diagnostic defects with other supporting evidence shown in Table 1. 

Evaluated secondary defects are represented by swollen cilia, cilia with vesicle formation between the membrane and axoneme, cilia with extra microtubules, compound cilia and the presence of free axonemes without membranes [27].

In our center, we calculate the deviation of angles using the NIS Elements software. The orientation of each cilium in the image was measured as the angle between some preset reference vector (0°) and the vector connecting microtubules in a central pair. The set of angles was created by measuring all of the cilia in one image. This data set was exported as a table to Microsoft Excel and preprocessed with Visual Basic for Applications (VBA) script before statistical testing. The script searches for a new reference vector for the given data set that after recalculating of all values has the minimal angle of 0° and the difference between minimal and maximal angles is the lowest possible for this data set. 

In order to investigate a large number of cilia and thus refine the quantitative analysis, we created a program that helps with counting and analyzing defects and classifying them into categories of cilia with normal ultrastructure and those with primary or secondary defects shown in Figure 2. Out of all primary defects listed in Table 1, PCD Quant is able to automatically analyze central complex defects and microtubular disorganization and differentiate them from secondary defects. The training set was composed of the following number of cilia: 1500 class 0 (normal), 400 class 1 (primary defects) 400 class 2 (secondary defects).

In the presented work, we compared the time consumption and accuracy of our original quantitative evaluation of ciliary ultrastructure using manual counting (with evaluation of the orientation of cilia) versus automatic analysis with the newly developing PCD Quant. The results were statistically evaluated. The classification performance was evaluated across ten images with more than 100 cilia each. The evaluation consisted of comparing points predicted by artificial intelligence, to the points manually annotated by a human expert. We used a one-versus-rest approach for multi-label classification to evaluate the metrics. Accuracy, sensitivity and specificity were used for each class to determine the predictive ability of the artificial intelligence for each class. The angles were measured by the software, then manually corrected. Both the values before the correction and after the correction were saved and used to evaluate the performance of angle prediction.

## 3. Results and Discussion

PCD Quant, an automatic analyzer, is able to analyze defects related to the presence of secondary changes. These include cilia that have membrane defects, free axonemes, or compound cilia. In terms of evaluation of primary defects, we initially focused on defects that are easier to detect at a magnification of 25,000×, which fits a larger number of cilia in one photo (up to about 100 cilia) and captures epithelial cells well. The program includes cilia with defective microtubule arrangements, with complex central pair defects and missing peripheral microtubules in the group of primary defects shown in Figure 2. Hallmark defects of the ciliary ultrastructure (Table 1), such as the absence of outer and inner dynein arms, cannot yet be evaluated using PCD Quant.

We bore in mind that this presented application should be able to be launched and executed easily in any operating system. Therefore, the graphical user interface (GUI) was implemented in Java and so the only operating system requirement was the installation of Java Runtime^®^. However, the computational backend of an application was implemented in Python^®^ and therefore necessary files were included in the install directory of the program. This means that the application is divided into two modules. The program installation file is publicly available on its GitHub repository website under releases (https://github.com/slobodaapl/pcd-gui/releases, accessed on 27 July 2021).

### 3.1. GUI—Image Selector and Viewer

The GUI of the application works as a controller for the backend computations carried out by the deep learning network. The main view was designed as an image viewer where the patient’s electron micrographs can be loaded and viewed before and after automatic evaluation. The images can be loaded by two common actions: drag-and-drop and using “Open Image Files” and the open image dialog window can select several images at once (Figure 2, #1). The loaded images are then shown in the left panel and by selecting them the image is displayed in the central interactive viewing pane. The displayed image can be analyzed individually by pressing the “Evaluate image” button (Figure 2, #2), which may be useful only if fast additional evaluation is needed. The sequential analysis of all loaded images can be started by pressing the “Evaluate Selected” button (Figure 2, #3).

### 3.2. Automatic Analysis—Deep Learning Model for Cilia Detection

One of our important contributions is the automatic quantitative analysis performed by a deep learning neural network model. We used the well-known architecture RetinaNet^®^ [28], which is used for object detection and classification achieving high accuracy on various detection tasks. The model is trained and optimized to detect the cilia in noisy microscopic TEM images and recognize whether they are intact or belong to defined categories of ciliary defects. The time necessary for the analysis of several images depends on the processing power of the CPU/GPU (central processing unit/graphics processing unit) and may take from seconds up to a minute per image. The current requirement and restriction for the method to be successful is the use of a particular microscope magnification. If the cilia size in the loaded image resolution is different than expected, the precision of the method is not guaranteed. Current training of the network was performed with a magnification of 25,000×.

The RetinaNet model uses the ResNet-18 [29] backend that utilizes a feature pyramid network (FPN) [30]. It accepts a preprocessed image that was inverted, de-noised, and split into 814 × 814p non-overlapping patches (image segments). During training, the training data were augmented by mirroring and rotation. The cilia on these patches were covered to balance classes’ distribution and to further augment data for the training. RetinaNet is an anchor-based approach detecting regions of interest and classifying them. It uses several scales of detection anchors for each class for potential variances in cilia size (detection regions). The network predicts the class for each anchor with a sigmoid scoring function in the output layer of the neural network, giving a score between <0 and 1>, independently for each scale and class. Non-maximum suppression filters duplicate and low-score detections, leaving only the most significant score. Displayed points over cilia are centroids of these bounding boxes, filtered further by score and proximity, where non-maximum suppression does not filter out overlapping bounding boxes.

The Adam [31] optimizer with a learning rate of 0.001 trains the network with mini-batches of 16 image segments for 28 epochs when training stops due to not improving validation loss in the last five epochs.

### 3.3. GUI—Classification Evaluation

The results of the automatic quantitative analysis are displayed again in the main view. The selected TEM image is now overlaid with annotation squares defining the location of the cilia and the color defining the predicted category. The quantitative analysis results are visible in the right top table containing counted numbers of normal cilia, cilia with primary defects and cilia with secondary defects. The lower list contains all predicted locations of cilia and the predicted classes. Unfortunately, all neural networks are only approximations of real data distributions, they cannot predict locations and classes correctly and suffer from statistical and training data errors. The list is important if the evaluator wishes to correct the classifications performed by the neural network model. The locations of cilia may be corrected by interacting with the image and square point annotations using mouse buttons in the view panel. The right button removes the annotation, and the left button has two uses, either it is possible to move the selected square point to correct the position of cilia or to create a new annotation point with a category selected in the drop-down menu “Point type” (Figure 2, #4).

### 3.4. Automatic Measurement of Angles

After the optional manual correction of possible misdetections and misclassifications, the application offers the automatic measurement of rotation angles of central pairs. This process is also fully automatic, and the user only has to start the analysis by pressing the “Calculate angles” button (Figure 2, #5). The process measures angles for all possible cilia and computes the average rotation and deviation from the corresponding angle. The individual angles are drawn on the visualized image for better visual interpretability and proof of measurement accuracy. Numerical values are displayed in the cilia list on the right, and the statistical values are displayed on top of the statistical table (Figure 2). If any of the measured angles seem to be wrong, the interface allows a user to correct individual angles by dragging the graphical angle lines manually. 

The angle measuring algorithm tries to locate the central pair in the detected cilia region (patch) with a determined size of 150 × 150 pixels containing whole cilia. Then, the correlations between cilia patch and preselected kernels with microtubule-like ellipses detect pixels with the highest correlation—regions with microtubules. After several morphological operations cleaning the noise, the contour and location analysis state the mass center of all detected microtubules, which should be close to one of the detected microtubule regions. These regions are then considered as central pairs. The angle is measured according to the significantly different pixel intensities of the detected central pair and computed from the bounding box of a rotated rectangle. 

Unfortunately, the noisy nature of TEM scans and the low-value differentiability of microtubules from the background presents a sometimes unsolvable barrier that deviates from the measurements. Therefore, in some cases, this problem inserts a small statistical error to the computation of the average angle, which may be ignored thanks to averaging of all angles and the wide problematic intervals of PCD. Sometimes the algorithm cannot find the central pair or selects the wrong region. In that case, the algorithm cannot measure the angle and returns the value −1 (instead of the actual angle). The default displayed value of the unmeasured angle is −0.

### 3.5. GUI—Export of Analysis

The application supports the easy exporting of .csv files for individual loaded images and one for joint statistics containing all the results of the analysis. 

### 3.6. Statistical Evaluation

Automatic analyzer of the ciliary ultrastructure is a tool that allows one to diagnose selected primary and secondary defects of the cilia in an electron microscope image. To validate the results, we selected 10 randomized samples from patients that had sufficient amounts of cilia in each photo (each above 100) and analyzed the defects using the cilia counting program normally used at our department in parallel with the automatic analysis. We saved the individual annotations and compared the individual evaluations as well as the speed of the individual analyses. The statistical evaluation of the results, accuracy, precision, sensitivity, and specificity [32] of the comparison of the automatic analysis with the manual evaluation is shown in Table 2.

In particular, the biggest issue represents the evaluation of primary defects, where the values of false positives are at the level of true positives. We assume that the low values of sensitivity and precision in the determination of primary defects are due to the insufficient amount of stored data. PCD is a rare disease, which significantly limits data collection and the creation of a representative set of images containing an adequate number of cilia. This fact limits the rapid progress of the deep learning network. We see another problem in the analysis options, which are limited to high-quality images with a sufficient number of cilia in cross sections with the TEM resolution set at 25,000× magnification. The PCD Quant cannot yet analyze images at a different magnification. The detection and classification of cilia independent of one fixed magnification could be improved by the utilization of more data and a training network on various magnifications, however, the measurement of angles might require a more sophisticated algorithm.

A precision-recall curve (Figure 3a) was used to measure the performance of the scoring function at various thresholds, with all contributing classes averaged. An ROC curve would not be viable here due to severe data imbalance and method of scoring. An F1 curve (Figure 3b) was constructed from the precision and recall scores at various scoring function thresholds to determine the best threshold for the classifier. The classifier performs well at a quite low threshold, but this also signifies low certainty of classification, mainly due to lack of data. A higher threshold (0.75) was used in PCD Quant to signify confident predictions, where predictions below this threshold are marked yellow in the table of cilia to be verified. Predictions with a threshold smaller than 0.15 were filtered out.

We also calculated Cohen’s kappa from the results. Including the background class misclassifications, the kappa evaluates to 0.613, and excluding the background class, it evaluates to 0.781, signifying moderate to strong data reliability [33]. This can be later improved with a larger data set.

Furthermore, we also evaluated the deviation of the central pair angles (Figure 4) on another randomized group of 10 patients. The results from comparing automatic analysis of the manual evaluation of angles of the individual deviations of the angles are listed in Table 3. 

The mean absolute error of the predicted angles against the corrected ground truth values evaluates to 11.198 degrees, and the standard deviation of the absolute errors of angle pairs is 33.522. The ratio of correctly predicted angles to all angles is 79.45%, where the predicted angle was determined to be correct if it fell within a 5 degree interval around the ground truth value. A Wilcox mean similarity test revealed samples of individual images not to be different (p > ~0.449) for the ground truth and predicted angles, signifying a certain degree of reliability in term of similarity. This is further indicated by the mean absolute percent error, which is 75.39%.

In terms of time, one manual examination looking at around 300 cilia using the NIS Elements program takes an average of 15 min of full concentration with continuous labeling of cilia. Automatic analysis evaluates the same number in an average of 3 min, which certainly simplifies the work.

A significant difference in the individual methods is due to the inaccuracy of the automatic analysis, which is still in the process of development. Evaluation of the individual images and saving the results were followed by editing the labeling in the program for automatic analysis according to accuracy and saving annotations for programmers who integrate the results into the program. Compared to other ciliary feature counting programs [34], automatic analysis lags significantly behind the division of cilia into individual defect subtypes. Individual subcategories of defects can be included in the program, shown in Figure 5. The added subcategories are evaluated manually with the storage of individual marks and later implementation into the program.

## 4. Conclusions

The PCD Quant, an automatic ciliary analysis program, cannot be used as a stand-alone analysis tool because it fails to assess class 1 hallmark defects associated with the diagnosis of primary ciliary dyskinesia, such as the missing outer dynein arms and inner dynein arms.

The program is also not yet able to evaluate separate subcategories of individual primary and secondary defects and puts them into three basic categories: normal, primary, secondary. However, it allows the addition of individual subcategories, the subsequent assignment of cilia with the saving of annotations and their gradual implementation into the program. By doing so, we expect a significant improvement in the automatic analysis with the possibility of involving the analysis of defects that the program has not yet been able to evaluate. 

Despite the impossibility of using PCD Quant as a stand-alone diagnostic tool, we see great potential for the future of automatic analysis. Therefore, we simultaneously ran the analysis automatically and by manual counting with constant comparison of results and storage of annotations. This program is our first version of an automatic analyzer in which we assume further development and classification of defects into other subgroups that would lead to more accurate diagnostics.

## Figures and Tables

**Figure 1 diagnostics-11-01363-f001:**
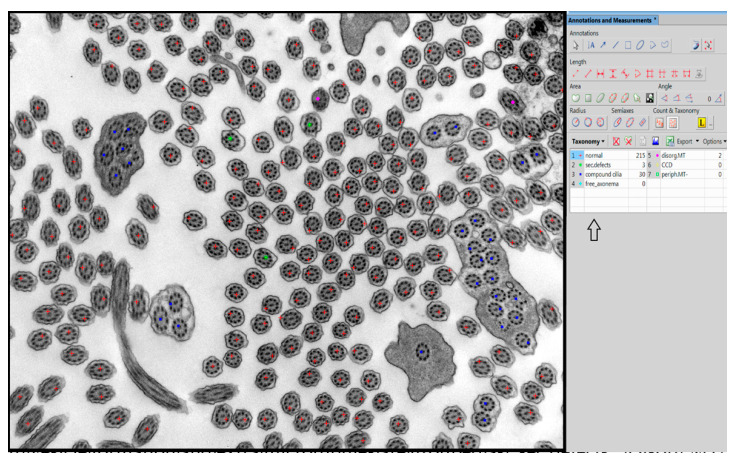
Example of manual evaluation using NIS Elements program with table in the right panel (arrow). Swollen cilia, cilia with vesicle formation and extra microtubules are grouped as “sec.defects”. Disorg.MT = disorganized microtubules, CCD = central pair complex defect, periph.MT- = missing peripheral microtubules. Compound cilia and free axonemes are categorized individually although they belong to secondary defects.

**Figure 2 diagnostics-11-01363-f002:**
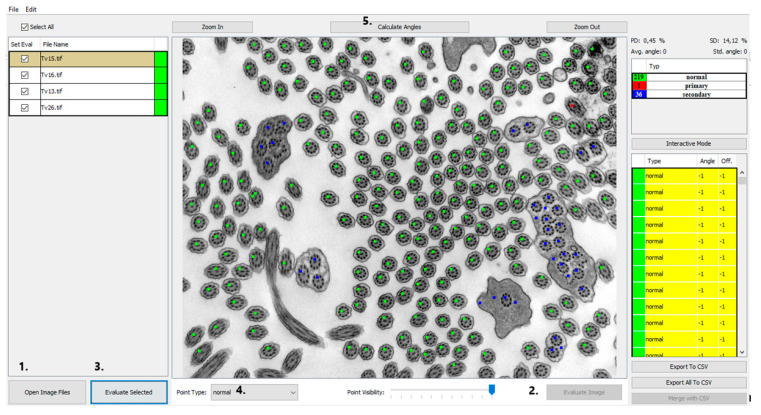
PCD Quant, the main window (the same photo as in Figure 1). This view presents a window displaying a list of loaded images in the left panel, color points on the detected cilia in the central image and the computed statistics in the right panel. The numbers direct to main control buttons: load images (**1.**), evaluate current image (**2.**), evaluate all selected images (**3.**), select annotation point (**4.**) and calculate angles (**5.**) This photo was selected as an example of how PCD Quant works without further editing. Compared to the manual evaluation, a difference can be seen in the number of cilia marked as “primary”, meaning primary defects. PCD Quant was able to detect one primary defect as opposed to two when manually evaluated in NIS Elements (Figure 1). A difference is also observable in detection of secondary defects (36 versus 33 in the manual evaluation).

**Figure 3 diagnostics-11-01363-f003:**
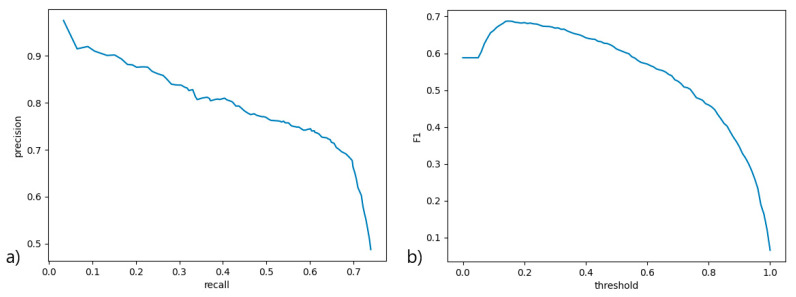
(**a**) The precision-recall (precision-sensitivity) curve signifying the ratio between the percentage of relevant detected samples to all samples and the percentage of correctly classified relevant samples, across various score thresholds; (**b**) the F1 score curve across various score thresholds to determine the most suitable threshold for filtering detected objects by score and demonstrating overall model performance.

**Figure 4 diagnostics-11-01363-f004:**
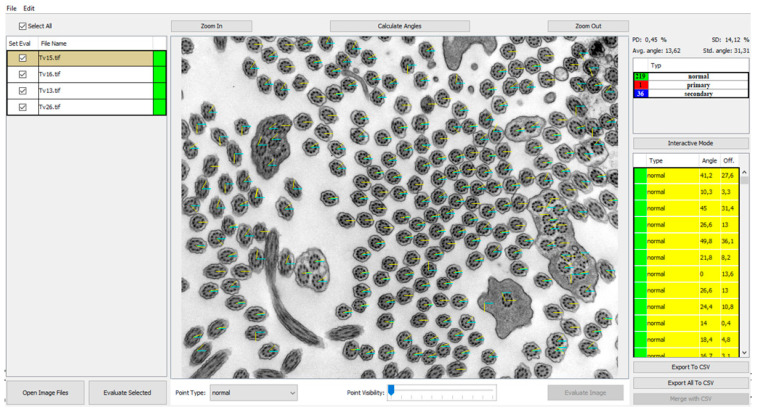
Program view of the central pair angles measured and visualized in the analyzed image with points visibility set to 0 (invisible). The base level of 0 angle is *X* axis represented by blue angle line. The yellow lines represent the orientation deviation from base level. Individual angle values are shown in the right panel listing angles and offsets from the mean.

**Figure 5 diagnostics-11-01363-f005:**
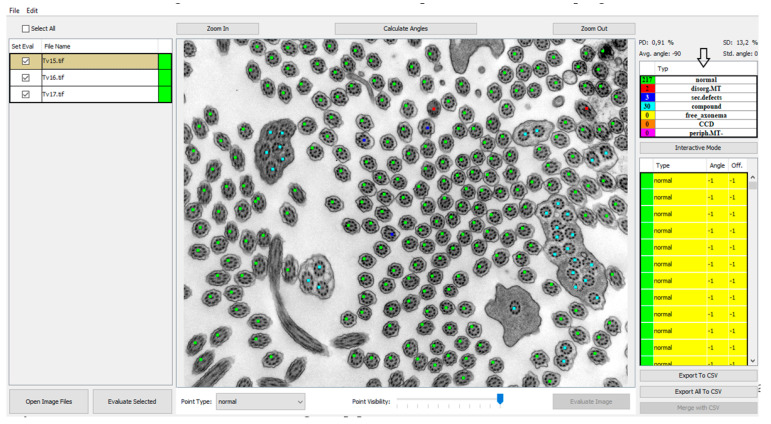
Image displays the possibility of adding other classes necessary for precise identification of primary and secondary defects. The classes are visible in the right top panel (arrow).

**Table 1 diagnostics-11-01363-t001:** International guidelines table classifying defects into class 1 and class 2 defects [20].

**Class 1 defects: Hallmark diagnostic defects**
Outer dynein arm defect
Outer and inner dynein arm defect
Microtubular disorganization and inner dynein arm defect
**Class 2 defects: Indicate a PCD diagnosis with other supporting evidence**
Central complex defect
Mislocalization of basal bodies with few or no cilia
Microtubular disorganization defect with inner dynein arm present
Outer dynein arm absence in 25–50% cross sections
Combined inner and outer dynein arm absence from 25–50% cross sections

**Table 2 diagnostics-11-01363-t002:** This table shows the accuracy, precision, sensitivity and specificity for each category (normal, primary defects, secondary defects) counted from list of true positives (TPs), false positives (FPs), true negatives (TNs), false negatives (FNs).

	Accuracy	Precision	Sensitivity	Specificity
normal	80.73%	89.89%	74.89%	88.62%
primary	79.30%	49.09%	43.37%	88.49%
secondary	81.31%	74.75%	71.53%	86.69%
overall	80.43%	77.09%	67.94%	87.90%
	**TP**	**FP**	**TN**	**FN**
normal	498	56	436	167
primary	108	112	861	141
secondary	299	101	658	119
overall	905	269	1955	427

**Table 3 diagnostics-11-01363-t003:** Statistics for angles: Mean absolute error (MAE) and its standard deviation (STD) for evaluated 10 patients comparing the automatically measured angle values against manual corrections.

	MAE	STD
**patient 1**	7.022	22.024
**patient 2**	14.484	28.271
**patient 3**	16.886	46.042
**patient 4**	13.385	48.093
**patient 5**	20.333	50.495
**patient 6**	20.068	30.246
**patient 7**	6.075	32.061
**patient 8**	4.921	19.984
**patient 9**	9.986	37.685
**patient 10**	8.613	34.651

## Data Availability

The data are not publicly available due to the presence of sensitive patient data.

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
