# Peer review of "Quantitative Assessment of Ciliary Ultrastructure with the Use of Automatic Analysis: PCD Quant"

_diagnostics, 2021, doi:10.3390/diagnostics11081363_

Round 1

Reviewer 1 Report

The manuscript by Felšöová et al. regards the field as extremely important for the whole PCD (Primary Ciliary Dyskinesia) community. PCD is a rare genetic disease, caused by the defects of motile cilia, small cellular projections present on many cells of the body, including the cells that form the lining of the respiratory airways. Defects of motile cilia cause the hallmark symptom of PCD - recurrent respiratory infections, which with time lead to serious malfunction of lungs and even to lung transplantation. Ciliary ultrastructure analysis using TEM is a current gold standard for the disease’s diagnostic pathway (Shoemark et al. 2020). The TEM analysis is a very tedious and time-consuming process, conducted manually by an experienced operator. To date, there is a lack of efficient tools enabling automatization of the diagnosis.

The manuscript is well written with a comprehensive introduction to the topic.  Results from this study are an important attemp to develop analytical software supporting the operator with the automated image analysis. The PCD Quant tool was designed to determine the ratio of primary and secondary defects in cilia ultrastructure as well as analysis of the mutual 28 orientation of cilia in the ciliary border.

The study presents interesting results, however, there several areas of significant concern outlined below.

  1. There is a lack of precise information on how the deep learning network was trained and test. The testing set was shown, but there is no information about the training set. Also, parameters applied for the model used in the experiment (RetinaNet® ) are missing.
  2. The authors can present more metrics related to the machine learning model performance like Kappa statistics, Rock Curve, and AUC.
  3. The most awaited functionality of the tool is the ability to assess class 1 hallmark defects associated with the diagnosis of PCD. However, it failed to achieve the required sensitivity and precision. I would suggest using a bigger data set or further optimization of model parameters to increase these metrics before the publication of these preliminary results.

Reviewer 2 Report

In my opinion and based on my personal experience, these types of tools are necessary to reduce the study times necessary for the correct analysis of samples that are of high complexity. My opinion is to accept this manuscript for publication, but unfortunately this cannot be done in its current form .

Comments:

  1. A better description of the methodology and algorithms used by the program would be necessary, as well as greater detail in the presentation of the results.
  2. In this article the authors included 10 PCD diagnosed patients. More detail regarding the diagnosis of such patients would be helpful. It would be advisable to include a table summarizing their clinical features in relation to age at diagnosis, onset of symptoms, clinical images, etc (you can find an example in Mata M, Lluch-Estellés J, Armengot M, Sarrión I, Carda C, Cortijo J. New adenylate kinase 7 (AK7) mutation in primary ciliary dyskinesia. Am J Rhinol Allergy. 2012 Jul-Aug;26(4):260-4. doi: 10.2500/ajra.2012.26.3784. PMID: 22801010.)
  3. For a correct diagnosis, it is necessary to evaluate at least 50 axonemes in cross section in the case of PCD and even more in the case of secondary dyskinesias. In the manuscript the authors analysed up to 100 cilia, but was it possible to find 100 cilia in cross section of all included patients? How many axonemes exactly did they count from each patient?.
  4. How does the program discriminate whether it is analyzing the proximal or distal portion of a cilium? this is very important in the case of certain structural alterations.
  5. How does the program determine if the membrane is intact or not?
  6. Could this program also identify central complex defects?
  7. The limitations observed by the authors could be due to the small size of the sample. It is true that PCD is a rare disease, but this question should be addressed in the discussion.

Finally, I want to acknowledge to the authors for the effort and I look forward to receiving the answers to my comments.

Reviewer 3 Report

A software allowing automatic analysis and classification of structural defects in a large number of cilia would be a welcome aid in PCD diagnostics. Unfortunately, the capacity of the software in its present form does not meet criteria needed to call it a diagnostic tool. The authors are aware of it and they rightly state that (line 316) “The PCD Quant as an automatic ciliary analysis program cannot yet be used as a stand-alone analysis tool because it fails to assess class 1 hallmark defects associated with the diagnosis of primary ciliary dyskinesia, such as missing outer dynein arms and inner dynein arms with missing radial spokes. The program is also not yet able to evaluate separate subcategories of individual primary and secondary defects and puts them into three basic categories: normal, primary, secondary…. we see great potential for the future in the automatic analysis and therefore we run the analysis carried out simultaneously automatically and by manual counting in the program with constant comparison of results and storage of annotations.“  Indeed, there is a potential for machine learning, as the program allows the addition of individual subcategories with saving annotations and their gradual implementation into the program.

However, statements like: (line 157) “This classification is our first version of an automatic analyzer in which we assume further development”; or (line 326) “...a fully automatic analysis of all primary and secondary defects, with the involvement of dynein arm analysis in the future” need further work and at least preliminary results to be shown to be more than assumptions.

The authors wrote: line 155/6 “…we created a program capable of automatically analyzing defects and classifying them into categories of cilia with normal ultrastructure and those with primary or secondary defects…” Line 138/9 “ In the overall assessment, we follow the latest international guidelines that classify ciliary defects into two classes [20]. This is a huge overstatement. The whole procedure described in the manuscript has little in common with defects classification into the classes described in [20]. In particular, there is no possibility to identify any of the class 1 defects. Central complex defect (one of the class 2 defects) would be the only feature, which could be classified at the magnification used by the authors.

In other words, the only capacity of the software the Authors present can be described as “distinguishing cilia with central complex defects from cilia with secondary defects (defects of a membrane, free axonemes, or compound cilia) and from the apparently normal cilia with no the central complex defects (whether they have dynein arms defects or no)”. Such an application may be useful to automatically screen samples for the large number of secondary effects.

Statistics presented in Table 2 is somewhat confusing. The reported number of 1100 cilia (line 256) used for the statistics is not consistent with the counts shown in Table 2; three rows, for normal, primary and secondary, should add up to 1100 each; the meaning of overall (adding up to 3556) is not clear.  

The authors explain low specificity and low precision (both derived from the ratio of true vs false primary positives) by the insufficient amount of stored data. While the statistics was based on the analysis of 10 individuals, the print screen from a single sample (Fig. 2) shows a single primary cross section (compared to three in manual assessment in Fig. 1.). It does not look compatible with the positive primary counts (true or false) presented in Table 2.

Automatic assessment of cilia angles appears a useful, time-saving feature, although the performance would need to be improved.  The sentence in line 277 is truncated.

The graphical unit interface is nice and clear; the description of the interface is correct and well-articulated. The remaining text requires a thorough revision by English-speaker, especially in parts concerning cilia biology. In its present form it is not only awkward but sometimes confusing or plain incorrect (some examples below).

(line 68) The importance of early and accurate diagnosis significantly affects the prognosis of patients … - grammar

(line 81) … software enhancing images of ciliary ultrastructure is currently developing – grammar

(line 92) …mutual orientation of cilia in the ciliary border – what is the meaning of “ciliary border”

(line 99) …endobronchial excision – meaning endobronchial biopsy?

(line 100) … from patients with symptoms suspected of primary ciliary dyskinesia - grammar

(line 101) … typical personal or family history, recurrent respiratory infection, bronchiectasis, hearing disorders, or right-sided organ placement – what is a typical personal or family history? hearing disorders or hearing impairment? right-sided organ placement – what if e.g. liver is left-sided?

(line 121) …we have so far used a manual taxonomic marking and counting of individual cilia - what is the meaning of taxonomic

(line 132) Evaluation of primary defects at a defined magnification ... is classified to subgroups  - grammar

(line 146) Orientation of each kinocilium in the image... The authors should check the definition of kinocilium and motile cilium

Round 2

Reviewer 1 Report

I am satisfied with the improvements. The explanations of the AI model and  metrics are sufficient. The lacking methodology was added. Also, the corrected conclusions emphasize that the program is still at the preliminary stage and needs further development and more samples to train algorithms. 

Reviewer 2 Report

The authors have answered all the questions raised in the first review of the article, which has improved the original manuscript, leaving all questions resolved. The only aspect that, from my point of view, should be implemented is the description of the clinical features of the patients recruited in this study. The diagnosis of PCD is highly complex and cannot be made without taking into account the clinical characteristics of the patients. That is why, in my opinion, it is very important to take these aspects into consideration. However, I leave it up to the editor to request or not such information.
Finally, I would like to congratulate the authors for the work they have done.